# Relationship Between Air Pollution and the Concentration of Nitric Oxide in the Exhaled Air (FeNO) in 8–9-Year-Old School Children in Krakow

**DOI:** 10.3390/ijerph18136690

**Published:** 2021-06-22

**Authors:** Marta Czubaj-Kowal, Ryszard Kurzawa, Henryk Mazurek, Michał Sokołowski, Teresa Friediger, Maciej Polak, Grzegorz Józef Nowicki

**Affiliations:** 1Department of Paediatrics, Stefan Żeromski Specialist Hospital in Krakow, Na Skarpie 66 Str., PL-31-913 Krakow, Poland; mjsokolowski@gmail.com; 2Department of Alergology and Pneumonology, Institute of Tuberculosis and Lung Disorders, Prof. Jana Rudnika 3B Str., PL-34-700 Rabka-Zdrój, Poland; ryszard.kurzawa@gmail.com; 3Department of Pneumonology and Cystic Fibrosis, Institute of Tuberculosis and Lung Disorders, Prof. Jana Rudnika 3B Str., PL-34-700 Rabka-Zdrój, Poland; hmazurek@igrabka.edu.pl; 4Faculty of Health, Catholic University in Ruzomberok, Námestie A. Hlinku 48 Str., SK-034 01 Ruzomberok, Slovakia; t.friediger@gmail.com; 5Department of Epidemiology and Population Studies, Jagiellonian University Medical College, Grzegórzecka 20 Str., PL-31-531 Krakow, Poland; maciej.1.polak@uj.edu.pl; 6Department of Family Medicine and Community Nursing, Medical University of Lublin, Staszica 6 Str., PL-20-081 Lublin, Poland; gnowicki84@gmail.com

**Keywords:** nitric oxide, FeNO, air pollution, particulate matter, children

## Abstract

The consequences of air pollution pose one of the most serious threats to human health, and especially impact children from large agglomerations. The measurement of nitric oxide concentration in exhaled air (FeNO) is a valuable biomarker in detecting and monitoring airway inflammation. However, only a few studies have assessed the relationship between FeNO and the level of air pollution. The study aims to estimate the concentration of FeNO in the population of children aged 8–9 attending the third grade of public primary schools in Krakow, as well as to determine the relationship between FeNO concentration and dust and gaseous air pollutants. The research included 4580 children aged 8–9 years who had two FeNO measurements in the winter–autumn and spring–summer periods. The degree of air pollution was obtained from the Regional Inspectorate of Environmental Protection in Krakow. The concentration of pollutants was obtained from three measurement stations located in different parts of the city. The FeNO results were related to air pollution parameters. The study showed weak but significant relationships between FeNO and air pollution parameters. The most significant positive correlations were found for CO8h (r = 0.1491, *p* < 0.001), C_6_H_6_ (r = 0.1420, *p* < 0.001), PM_10_ (r = 0.1054, *p* < 0.001) and PM_2.5_ (r = 0.1112, *p* < 0.001). We suggest that particulate and gaseous air pollutants impact FeNO concentration in children aged 8–9 years. More research is needed to assess the impact of air pollution on FeNO concentration in children. The results of such studies could help to explain the increase in the number of allergic and respiratory diseases seen in children in recent decades.

## 1. Introduction

Air pollution is one of the most severe global threats in the modern world and impacts, in particular, large urban agglomerations. The World Health Organisation (WHO) reports that 9 out of 10 people in the world breathe polluted air, causing about 7 million deaths per year. In 2016, air pollution led to 4.2 million deaths. During smog alarms, the number of hospitalised patients with myocardial infarction (by 12%), strokes (by 16%) and respiratory diseases is increasing. According to WHO, European Union (EU) and UNICEF data, Poland, including Krakow, is one of countries with the most polluted air in the EU; in 2016, 33 Polish cities were among the 50 most polluted in Europe. The most frequently monitored components of air pollution in Poland are particulate matter (PM_10_, PM_2.5_), carbon monoxide (CO), nitric oxide (NO), and benzene (C_6_H_6_) [1,2,3,4,5,6,7].

Due to the developmental period, immaturity of the respiratory and immunological system, and detoxification mechanisms, children are most vulnerable to the penetration and concentration of air pollutants in the respiratory tract and to their toxic effects [8,9,10,11,12,13,14,15,16,17]. It is estimated that over 90% of children breathe toxic air every day, 300 million children live in areas with air pollution exceeding 6 times the standard, and 1.7 million children die due to air pollution, including one in 10 children under 5 years of age. Respiratory diseases, including asthma and its exacerbation in school-age children, especially those living in large urban agglomerations with a high level of air pollution, are one of the main causes of school absences, admissions to doctor’s offices and hospitalisation [1,2,3,4,18,19]. A simple method is needed to assess the impact of air pollution on the inflammation of the airways in children.

The nitric oxide concentration in exhaled air (FeNO), in recent years, due to the simplicity, safety, and non-invasiveness of the FeNO measurement, is increasingly used in the diagnostics of respiratory diseases in children [20,21,22,23,24,25]. It can be repeated many times in cooperating children. It can be done even in schools and does not require a medical laboratory or special preparation. The result is available immediately, which allows for a quick diagnosis and treatment. Elevated concentrations of nitric oxide in the exhaled air suggest eosinophilic inflammation of the airways. Numerous studies confirm the usefulness of FeNO measurements in the diagnostics, monitoring and treatment of asthma. Some studies suggest its usefulness in diagnostics of allergic rhinitis, chronic cough, eyelash discolouration and COPD [26,27,28,29,30,31,32,33,34,35,36,37,38,39,40,41,42,43].

However, only a few studies have assessed the impact of air pollution on the result of FeNO measurement in children. Most of these works were based on small patient groups, and only a few reports covered a large group of children [44,45,46,47,48,49]. As an example, in the study of the paediatric population, Koenig et al. [45] presented 19 children, Delfino et al. [48] 45 children and Permaul et al. [49] 271 children. Only a few researchers examined larger groups of children—Berhane et al. [47] 1211, Idavian et al. [46] 1326 and Zhang et al. [44] 3607. Nevertheless, the relationship between air pollutants and FeNO concentration is not fully understood and documented. Most studies on the effects of exposure to air pollutants related to the increase in FeNO concentration come from Western and Northern Europe [46] and North America [45,47,48,49]. Poland is characterized by much higher levels of air pollution, especially during winter [50]. The study aims to estimate the concentration of FeNO in the population of children aged 8–9 attending the third grade of public primary schools in Krakow, as well as to determine the relationship between FeNO concentration and dust and gaseous air pollutants.

## 2. Materials and Methods

### 2.1. Study Design and Participants

This population-based study was conducted across all public primary schools within the city of Krakow (*n* = 119). Children aged 8–9 attending the third grade of primary school were included in the investigation. During the research period, in the 2017–2018 school year, 7752 students participated in the third grade of public primary schools in the city of Krakow. The study consisted of two stages (Figure 1). Stage I of the study was carried out during the autumn and winter period, i.e., in the months from October 2017 to January 2018. A total of 5460 children participated in Stage I. Stage II covered the spring and summer months, i.e., May and June 2018, with 4580 children participating. The analyses included 4580 children participating in Stage I and Stage II of the study, which accounted for 83.9% of children examined in Stage I. The study group accounted for 59.1% of all children attending the third grade of public primary schools in the city of Krakow.

The data collection team included specially trained physicians. They conducted two meetings with the children in each class on two different days. On the first day, the children were given written information explaining the purpose, method of examination, and instructions on how to prepare for the examination. A form for parents/legal guardians to complete, which granted consent for the child’s participation in the study, was placed on a separate sheet of paper. The children were asked to present this information to their parents/legal guardians. The information package for parents/legal guardians included e-mail addresses and telephone numbers of the researchers who designed the study, so that parents/legal guardians could easily get in touch and clarify any doubts. On the second day, consent forms signed by the parents/legal guardians were collected from the children. An appropriate examination was then performed with only those children who had provided the signed consent of a parent or legal guardian. The research was carried out in the offices of the school nurse. Both healthy children and children with diagnosed respiratory diseases were included in the study. The presented material is a part of the research project conducted by the Department of Paediatrics, Stefan Żeromski Specialist Hospital in Krakow, Poland, as a part of the campaign to improve air quality, “Let’s be together in the fight for clean air in Krakow”, conducted by the Municipality of Krakow. The study was approved by the Bioethics Committee Institute of Tuberculosis and Lung Diseases, KB-26/2018.

### 2.2. Measurement of FeNO

The FeNO measurements were taken, according to the recommendation of the American Thoracic Society (ATS) [35,36,50] using the on-line method, with the Hyp’AirFeNO electrochemical analyser (MediSoft, Sorinnes, Belgium). This analyser guaranteed repeatable measurements of FeNO in the range of 0–600 ppb and did not require external calibration. The electrochemical breath analyser converts gas concentration into electrical signals. Patients exhaled through disposable mouthpieces at a constant flow of 50 mL/s for 6 s. The device was calibrated and used according to the manufacturer’s instructions and implemented in conjunction with a computer. The ExpAir software provided automatic recording of the study, automatic interpretation of results, data logging and a report. This device is easy to use, semi-portable, fast and cheap, compared to other analysers based on the chemiluminescent and laser method. It is characterised by good reliability and repeatability as well as a high level of cooperation with patients, especially children. The test was painless and non-invasive for the paediatric participants.

Two hours prior to testing, the children were to refrain from eating and drinking as well as physical activity, and not to be exposed to tobacco smoke. The tests were carried out between 8:00 a.m. and 2:00 p.m. Before the measurements, the children were instructed on how to perform the test, and the children’s ability to perform the procedure was checked. The FeNO measurement was performed twice for each patient. The first measurement was to teach the child how to perform the test, and the second was a diagnostic measurement; this result was retained for subsequent analysis. The FeNO result was presented in ppb values (parts per billion, 10^−9^), which is a dimensionless description of the ratio of two values. The upper limit or normal value range of the FeNO in children was set at 20 ppb [21,35,36,51].

Concentration values of FeNO above the accepted norm were divided into 3 groups, according to the concentration levels, reflecting the intensity of eosinophilic airways inflammation: Group I, NO 21–50 ppb; Group II, NO 51–99 ppb;and Group III, NO ≥ 100 ppb [21,28,36,47]. Those intervals corresponded, respectively, to mild, significant and severe eosinophilic inflammation [33,46].

### 2.3. Information on Concentration of Ambient Air Pollution

FeNO levels were correlated with the concentration of ambient air pollution PM_10_, PM_2.5_, and NO, CO, C_6_H_6_. The degree of air pollution was obtained from the Regional Inspectorate of Environmental Protection in Krakow (http://monitoring.krakow.pios.gov.pl) (accessed on from October 2017 to January 2018 and from May to June 2018). The concentration of pollutants was obtained from 3 measurement stations located in different parts of the city. The concentration values of the studied pollutants were obtained from the monitoring station located closest to the school where the FeNO measurement was performed. The average daily concentration of the analysed contaminants and the median from 7 days preceding the FeNO measurement, which best described the concentration distribution, were taken for analysis.

### 2.4. Statistical Analysis

Continuous variables were presented as means with standard deviation (SD) if normally distributed or median with first quartile (Q1) and third quartile (Q3). The Shapiro–Wilk test was used to assess conformity with a normal distribution. Categorical variables were reported as absolute numbers and percentages. Differences between groups were assessed with the Chi-square test for categorical variables, and the Mann–Whitney U, t test and Kruskal–Wallis test for continuous variables. Sperman’s rank correlations were used to investigate the relationships between FeNO and PM_10_, PM_2.5_, NO, CO, and C_6_H_6_. Moreover, the linear regression was used to consider the influence of age and sexto the relationships between air pollution and FeNO. The analysis of results is presented as a standardized beta coefficient with 95% confidence intervals (CI). Due to the right-skewed distribution of FeNO, the logarithm transformation was applied. Statistical analyses were conducted using IBM Corp. Released 2017. IBM SPSS Statistics for Windows, Version 25.0. Armonk, NY: IBM Corp. Statistical significance was set at the level *p* < 0.05.

## 3. Results

### 3.1. Characteristic of Participants

The age of the children was 8–9 years, average 8.9 (SD = 0.48). There were slightly more girls than boys in the study group—2753 (50.4%).

### 3.2. Distribution of Obtained FeNO Measurements

Table 1 shows the distribution of the obtained FeNO measurements in specific months of the study. In the first measurements, performed in Stage I (autumn–winter period), the FeNO range was from 2 to 144 ppb, median 11 ppb (Q1 = 8, Q3 = 17). The FeNO range in the second Stage II (spring–summer period) was from 1 to 196 ppb, median 12 ppb (Q1 = 8, Q3 = 18). There were no significant differences in the distribution of FeNO and categories of FeNO between the months in Stage I and Stage II. No significant correlation was observed between the age of the studied children and FeNO values at Stage I (r = 0.02, *p* = 0.14). Higher FeNO values were found in boys as compared to girls in both Stages (Stage I: boys median 12, Q1 = 8, Q3 = 18; girls median 11, Q1 = 8, Q3 = 16, *p* = 0.003; Stage II: boys median 12, Q1= 9, Q3 = 19, girls median 11, Q1 = 8, Q3 = 17, *p* = 0.002).

### 3.3. The Level of Air Pollution on The Days of FeNO Measurements

The level of air pollution in the range: particulate matter (PM) in two size fractions PM_10_, PM_2.5_, nitric oxide (NO), benzene (C_6_H_6_) and carbon monoxide (CO) on the days of FeNO measurements of Stage I and II is shown in Table 2.

### 3.4. Relationship Between FeNO and Air Pollution

Table 3 shows the relationship between the FeNO and the air pollution measured on the day of measurement, and the median value from the week before. We observed a significant positive relationship between FeNO and the value of air pollutants PM_10_, PM_2.5_, NO, CO and C_6_H_6_ in Stage I and for NO, CO and PM_2.5_ (only for concentration from the day of the FeNO test) in Stage II. In both Stages correlations of FeNO results with pollutants concentration were stronger for measurement from the day of examination than for median from the last 7 days. The strongest relationships were found for parameters CO8h (r = 0.149; *p* < 0.001), C_6_H_6_ (r = 0.143; *p* < 0.001) and PM_2.5_ (r = 0.106; *p* < 0.001) in Stage I. After adjustment for age andsex, the results were similar to those obtained in the univariable analysis and remained significant. The results of the multivariable linear regression are presented in Appendix A Appendix A.

## 4. Discussion

Our analysis of FeNO measurements made in a large group of children in Krakow showed a correlation between FeNO values and concentrations of air pollutants. The most significant positive correlations occurred for CO8h, C_6_H_6_, PM10 and PM2.5. Higher values of these parameters were associated with higher FeNO concentration values. It concerns both the concentration of air pollution parameters on the day of FeNO measurements and the median from 7 days before measurements were taken. The relationship described above was stronger in the autumn and winter period.

FeNO is a non-invasive biomarker with good repeatability of measurements, high specificity and sensitivity in diagnosing, monitoring or predicting responses to respiratory diseases [52]. Several epidemiological studies suggest a change in FeNO in the assessment of airway inflammation in association with air pollution, thus suggesting that short-term or long-term exposure to air pollution may be associated with alteration in FeNO. However, the research results remain inconsistent. Zhang et al. [53] found that an increase in PM_2.5_ by 10 μg/m^3^ was associated with an increase in FeNO by 2.59%. In studies by Zhao et al. [54], carried out in a healthy population, it was found that an increase in NO_2_ of 10 μg/m^3^ was associated with a 10.58% increase in FeNO. Similar results were reported in the population of people with COPD. Conversely, several studies showed that air pollution was not related to FeNO levels [55,56,57,58]. However, a meta-analysis of 27 studies by Chen et al. [59] on the impact of air pollution on FeNO levels in adults and children revealed that an increase in short-term exposure to PM_10_, PM_2.5_, NO2 and SO2 by 10 μg/m^3^ was associated with an increase in FeNO concentration by 3.2%, 2.25%, 4,9% and 8.28% (95% CI: 3.61%, 12.59%). Moreover, another meta-analysis by Khreis et al. showed that the onset of asthma was significantly associated with an increase in PM_2.5_ concentration by 1 μg/m^3^ in the air, by 2 μg/m^3^ for PM_10_ and by 4 μg/m^3^ for NO2.

During our own research during the autumn–winter period, we observed significant positive relationships between FeNO and air pollution parameters PM_10_, PM_2.5_, NO, CO8h, and C_6_H_6_. In the spring–summer period, air pollution in Krakow was lower than in the autumn–winter period, and we observed weaker relationships between FeNO and air pollution parameters. Significant positive relationships were found only in the case of CO8h, NO for the concentration on the day of the FeNO measurement and for the medians 7 days before the test, and PM_2.5_ for the measurement on that day. Higher values of FeNO measured at Phase II compared to Phase I despite the reduced exposure to air pollutants may be related to high exposure to pollens in the spring–summer period. However, this aspect was not analysed in this study. A stronger correlation was found between the FeNO concentration and the median PM_2.5_ concentration from the last 7 days than in the case of PM10, which is consistent with the results of the studies by MacIntyre et al. [60] who concluded that PM_2.5_ is possibly more harmful to the respiratory system than PM_10_. The same result was obtained by Zhang et al. [61] in a group of five-year-old children.

Most of the published studies on the impact of air pollutants on FeNO were completed in different study settings and in small groups of children—often selected by health status. Steerenberg et al. [62] were the first to describe a longitudinal observation of 82 children exposed to air pollution. Short-term exposure resulted in increases in FeNO levels, increased inflammatory, nasal markers and PEF changes. Delfino et al. [48] in a study of 45 schoolchildren with asthma showed a positive correlation between the two-day mean concentration of pollutants and FeNO. In two-pollutant models, the most significant correlation was found for elemental carbon (EC) and NO_2_. Koenig et al. [45] published a work on a small group of 19 children with asthma, aged 6–13 years. They demonstrated a relationship between FeNO and PM_2.5_ exposure in both winter and summer, with 10-day measurements taken at home or outdoors, from personal monitors and three central-site monitors.

Only a few studies included large groups of children. Berhane et al. [47] analysed the results of 2240 schoolchildren from Southern California, aged 7–9 years. They showed that short-term increases in air pollutant concentrations (PM_2.5_, PM_10_ and ozone (O_3_)) were significantly related to higher FeNO levels—independently of asthma and allergy status. They suggested that air pollution increases nitrosative stress in both healthy and atopic children. Idavain et al. [46] studied a group of 1326 schoolchildren from Estonia aged 8–12 years, living near oil processing factories. In a cross-sectional study, they showed that children exposed to higher levels of industry-specific air pollutants had significantly increased prevalence of high values of FeNO, respiratory symptoms and asthma. Children exposed to higher concentrations of C_6_H_6_ had a higher odds ratio (OR) of having FeNO levels ≥ 30 ppm.

In the presented own research, no correlation was observed between the age of the examined children and FeNO values in the autumn–winter period, and a weak correlation was observed in the spring–summer period. It could be explained by the narrow age range of the examined children, which was very similar. In both study periods, autumn–winter and spring–summer, we observed higher FeNO values in boys than in girls. There is controversy in the literature about sex differences in FeNO: similar to us, most researchers observed higher values in boys [25,41], but in other studies, the difference was not significant [24,44].

There are several potential mechanisms explaining the relationship between air pollution and FeNO concentration. In the case of PM, the mechanism of changes in the lungs can be explained as follows. Nitric oxide is produced by many cells and is generated through the amino acid L-arginine by nitric oxide synthase (NOS) [63]. NOS exists in three different isoforms: neuronal (nNOS), induced (iNOS) and endothelial (eNOS). Of the above-mentioned forms, iNOS appears to be a pro-inflammatory mediator and is associated with inflammatory diseases of the respiratory tract. Inhaled PM may penetrate the lining of epithelial cells, increasing L-arginine oxidation and activating lipid peroxidation [52]. This process influences the increased production of NO under oxidative stress conditions, which secondarily generates strong oxidation reactions [63,64]. The mechanism underlying the impact of gaseous pollutants in the air on FeNO concentration is probably modulated by DNA methylation in the arginase–nitric oxide synthesis pathway [65]. Studies on human vascular endothelial cells showed that hypomethylation of NOS2A can lead to an increase in FeNO by increasing the concentration of iNOS. Moreover, ARG2 hypomethylation may also increase FeNO levels by increasing the availability of L-arginase [66].

Growing evidence suggests a link between ozone (O_3_) concentration and respiratory disease [67,68], but it is the major photochemical pollutant and strong oxidant that has not attracted the attention of many researchers. Several animal studies investigated the relationship between ozone exposure and FeNO. Niu et al. [69] showed that ozone can cause a decrease in NOS2A methylation and increase induced NOS expression, suggesting that inhalation of ozone may affect DNA methyltransferases. The authors hypothesized that increased FeNOlevels are also associated with decreased arginase concentration and increased arginase-2 methylation. Studies on O3 exposure and increased FeNO concentration in children and adolescents produced different results. Barraza-Villarreal et al. [70], Nickmilder et al. [71], and Karakatsani et al. [72] observed a positive relationship between ozone exposure and FeNO. On the other hand, Altuğ et al. [73] and Barath et al. [74] did not confirm the influence of O_3_ on the FeNO level. Our research did not assess the impact of O_3_ on the FeNO level because only one measuring station within the city of Krakow (located at Bujaka Street) provides information on the concentration of O_3_.

The study of FeNO and its correlation with air pollution is of clinical importance for the early detection of eosinophilic airway inflammation, especially in children living in highly polluted areas. It allows selecting a group of children with inflammatory lesions and the risk of damaging the respiratory tract that requires further diagnostics. FeNO measurement is simple, cheap, non-invasive, and easy to perform, even for the youngest children. The assessment of the correlation of FeNO with air pollution draws attention to the impact of environmental pollution on the health of children. It prompts taking actions to protect the environment, which translates into the improvement of the health of the society. The clinical importance of FeNO measurement in the detection of eosinophilic inflammation of the airways, diagnosis, treatment, and its modification in bronchial asthma [26,27,28,29,30,31,32,33,34,35,36,37,38,39,40,41,42,43], as well as correlation with air pollution, is confirmed by many authors [44,45,46,47,48,49]. FeNO correlates well with other indicators of eosinophilic inflammation assessed, e.g., in the biopsy material, bronchoalveolar lavage fluid, or induced sputum [21,28]. However, these methods are invasive, time-consuming, expensive, and difficult to perform in children. As a population-level study, our study was designed to demonstrate the relationship of FeNO levels with air pollution. Further studies are needed to evaluate the correlation between FeNO and air pollutants, taking into account individual measurements of exposure to air pollution and considering confounding variables to more accurately demonstrate the relationship and utility at the clinical level. Allergic diseases, asthma, allergic rhinitis, atopic dermatitis, exposure to animal and pollen allergens, exposure to tobacco smoke, infection frequency, distance from pollutant emission sources, age, gender, chronic diseases, and immunization should be taken into account—these are socioeconomic factors.

It is worth considering the strengths and weaknesses of this study. An extensive analysis of the literature suggests that this is the first study in which such a large group of children in Poland participated. Secondly, the study included 4580 children living in the second largest city in Poland, which is characterised by some of the most severe pollution in Poland and Europe [1,2,3]. Thirdly, the study covered a group of children attending public primary schools in the city of Krakow, and this city is characterised by the most polluted air in Poland and Europe.

Nevertheless, this study has several limitations. First, it is a cross-sectional study and, therefore, shows no cause-effect or time-effect relationship between FeNO concentration and air pollution. Secondly, only the basic sociodemographic data of the surveyed children (i.e., sex and age) were considered. We did not gather any information on the influence of other sociodemographic variables, such as lifestyle (e.g., exposure to tobacco smoke), the occurrence of respiratory diseases (such as bronchial asthma, atopy, inhalation allergies, etc.) and the medications taken; these features may influence the relationship between air pollution and FeNO concentration. Therefore, no detailed analyses of the influence of these variables on the FeNO concentration were performed. These additional factors, as well as allergy to trees or grass pollens and possible asthma exacerbations probably had an impact on the higher FeNO values during spring–summer in some of the examined children; however, passive exposure to tobacco smoke may lead to a decrease in FeNO concentration [75,76]. Third, the information on air pollution (obtained from air quality monitoring stations) may not match the actual air pollution exposure for a particular child. The assumed values of exposure levels to air pollution could cause the relationships to be biased. However, the natural topography of the basin in which Krakow is located results in accumulation and maintenance of the level of air pollution in Krakow with relatively uniform distribution within the city [77,78]. Moreover, in order for the air pollution monitoring data to best reflect the actual exposure to pollutants of the studied children, in the exposure assessment, we chose the monitoring stations closest to the school where we conducted the research (mean distance was equal to 3.1 km (*SD* = 1.5)).

## 5. Conclusions

Air pollution positively correlates with FeNO concentration in children aged 8–9 years. Although in our research this correlation was weak, it was important for dust and gaseous pollutants. More research is needed to assess the impact of air pollution on FeNO concentration, taking into account confounding variables, such as the sociodemographic and medical variables of respondents. The results of such studies could help explain the increase in the number of allergic and respiratory diseases seen in children in recent decades.

## Figures and Tables

**Figure 1 ijerph-18-06690-f001:**
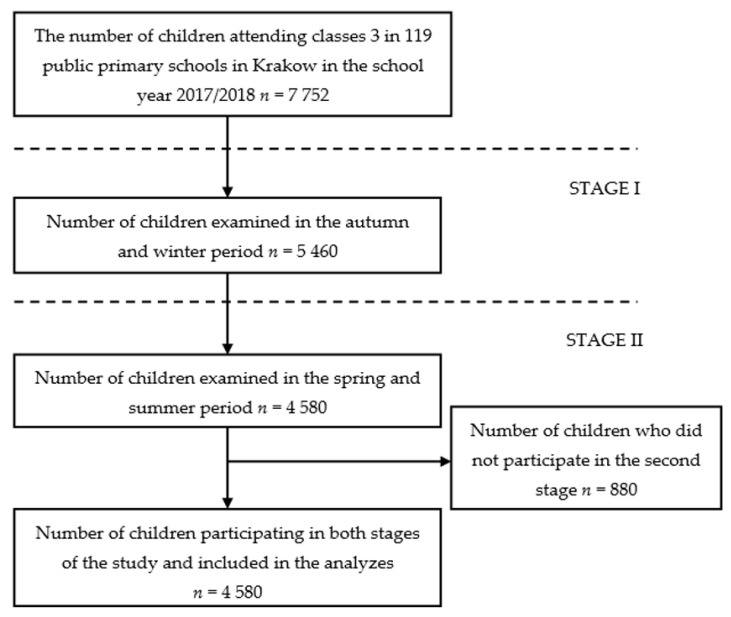
A flow-chart demonstrating the selection of studied groups.

**Table 1 ijerph-18-06690-t001:** The distribution of the measured values of FeNO.

FeNO (ppb)								
Month	*n*	Me (Q1–Q3); Min.–Max	*p*	0–20 *n* (%)	21–50 *n* (%)	51–99 *n* (%)	≥100 *n* (%)	*p*
Stage I (autumn–winter period):								
October	191	11 (8–15); 2–104	0.14 ^A^	165 (86.4)	22 (11.5)	3 (1.6)	1 (0.5)	0.068 ^C^
November	1883	11 (7–17); 2–136		1545 (82.1)	292 (15.5)	40 (2.1)	6 (0.3)	
December	1673	12 (8–17); 2–144		1374 (82.3)	231 (13.8)	56 (3.1)	12 (0.8)	
January	833	11 (8–16); 4–125		704 (84.5)	104 (12.5)	23 (2.6)	2 (0.4)	
Stage II (springer–summer period):								
May	2279	12 (9–18); 1–154	0.23 ^B^	1832 (80.4)	365 (16.0)	74 (3.3)	8 (0.4)	0.15 ^C^
June	2301	11 (8–18); 1–196		1796 (78.0)	400 (17.4)	98 (4.3)	7 (0.3)	

Me: median; Q1: lower quartile; Q3: upper quartile; ^A^ *p*-values calculated using Kruskal–Wallis test; ^B^ *p*-values calculated using Mann–Whitney test; ^C^
*p*-values calculated using chi-square.

**Table 2 ijerph-18-06690-t002:** The value of air pollution during the period under investigation.

Stage/Month	NO (µg/m^3^)	CO8h (µg/m^3^)	C_6_H_6_ (µg/m^3^)	PM_10_ (µg/m^3^)	PM_2.5_ (µg/m^3^)
Stage I (autumn–winter period):					
	45 (14–91); 1–331	1063 (744–1501); 343–3250	2 (1.2–3.9); 0.4–25.4	43 (26–67); 7–192	31 (18–51); 4–163
October	32 (14–78); 1–181	834 (658–1093); 353–1588	1 (1.0–2.2); 0.6–3.5	30 (21–50); 7–107	22 (13–37) 4–79
November	55 (20–115); 2–278	1247 (919–1907.5); 365–3080	3 (1.6–4.0) 0.7–5.4	48 (29–75); 13–127	34 (22–57); 7–93
December	45 (14–83); 2–202	1072 (772–1498); 343–2016	2 (1.3–3.1); 0.5–7.0	44 (29–65); 12–121	30 (20–50); 5–111
January	38 (10–107); 1–331	1202 (820–1624); 467–3250	3 (1.9–5.5); 0.4–25.4	53 (32–76); 13–192	40 (21–62); 7–163
Stage II (spring–summer period):					
	12 (4–47); 0–108	627 (527–741); 234–1185	1 (0.4–1.4); 0.2–2.9	28 (22–37); 9–59	17 (13–21); 4–39
May	13 (4–41); 0–99	666 (532–741); 234–899	1 (0.5–1.7); 0.2–2.9	30 (24–40); 12–59	17 (13–21); 7–39
June	8 (4–50); 0–108	612 (475–724); 330–1185	1 (0.4–1.2); 0.2–2.1	27 (22–35); 9–52	16 (4–35); 11–21

Date are presented as: Me (Q1–Q3); Min.–Max.

**Table 3 ijerph-18-06690-t003:** Relationship between FeNO and air pollution in the study group.

FeNO	Stage I (Autumn–Winter Period):		Stage II (Spring–Summer Period):	
	r	*p*	r	*p*
The day of measurement:				
NO (µg/m^3^)	0.053	<0.001	0.031	0.035
CO8h (µg/m^3^)	0.149	<0.001	0.067	<0.001
C_6_H_6_ (µg/m^3^)	0.143	<0.001	0.016	0.271
PM_10_ (µg/m^3^)	0.097	<0.001	0.027	0.065
PM_2.5_ (µg/m^3^)	0.106	<0.001	0.039	0.022
The median value from the week before:				
NO (µg/m^3^)	0.028	0.016	0.051	<0.001
CO8h (µg/m^3^)	0.057	<0.001	0.056	0.001
C_6_H_6_ (µg/m^3^)	0.043	0.018	0.024	0.106
PM_10_ (µg/m^3^)	0.053	<0.001	0.012	0.417
PM_2.5_ (µg/m^3^)	0.044	<0.001	0.002	0.899

## Data Availability

The data are available from the corresponding author upon reasonable request.

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
