# Peer review of "Relationship Between Air Pollution and the Concentration of Nitric Oxide in the Exhaled Air (FeNO) in 8–9-Year-Old School Children in Krakow"

_ijerph, 2021, doi:10.3390/ijerph18136690_

Round 1

Reviewer 1 Report

The study found significant correlations between air pollutants measured near schools and FeNO of children attending those schools with a large sample size. But the manuscript might need some revisions to demonstrate the observed associations are not biased and articulate the contribution of this study to the literature.

  1. The associations between air pollutants and FeNO were assessed using correlation. The analysis with adjustment in the supplemental material only adjusted for age and sex. There are many other confounders that can confound the observed correlations. Even if the individual-level SES was not collected, what about the neighborhood-level SES? The studied children included both healthy and the ones with diagnosed respiratory diseases. Would disease status modify the correlation between air pollution and FeNO? Would diagnosis of respiratory diseases indicate overall worse environmental quality (air, water, soil pollutants, less green space etc.) or worse healthcare in the neighborhood, which can confound the correlation between air pollution and FeNO?
  2. Discussion is needed regarding the impact of exposure misclassification on the observed correlations, e.g., exposures at home. Consider reporting the distances between the nearest monitors and the studied schools.
  3. It is unclear what is the goal for the analysis on the correlation of FeNO between Stage I and Stage II (section 3.4). Can the authors further elaborate?
  4. Can the authors comment on the clinical significance of the results? Based on the linear regression results, it seems that the increase in FeNO was less than 1% for an SD increase in air pollutants. Given the range of FeNO in the population and in each clinical category, can the authors comment on the clinical significance of the findings?
  5. The authors listed the sample size of many previous studies on this topic in the introduction. If these studies showed consistent results, small sample size might not be a sufficient reason for an additional study with a larger sample size, especially if other studies with smaller sample size had more information on potentially important confounders. Can the authors elaborate the gaps in the literature and what is the contribution of this study?

Author Response

Response to Reviewer 1 Comments

Respected Reviewer,

We wish to submit a revised manuscript “Relationship between air pollution and the concentration of nitric oxide in the exhaled air (FeNO) in 8-9-year-old school children in Krakow” (Manuscript ID: ijerph-1226873) for further consideration by the International Journal of Environmental Research and Public Health.

We do confirm that we have taken all the valuable reviewers comments under consideration and applied changes to the manuscript. The improvements are marked in blue in the manuscript.

Point 1: The study found significant correlations between air pollutants measured near schools and FeNO of children attending those schools with a large sample size. But the manuscript might need some revisions to demonstrate the observed associations are not biased and articulate the contribution of this study to the literature.

Response 1: Respected Reviewer, Thank you very much for your review. We appreciate your suggestions and comments, which will undoubtedly improve the quality of our manuscript.

Point 2: The associations between air pollutants and FeNO were assessed using correlation. The analysis with adjustment in the supplemental material only adjusted for age and sex. There are many other confounders that can confound the observed correlations. Even if the individual-level SES was not collected, what about the neighborhood-level SES? The studied children included both healthy and the ones with diagnosed respiratory diseases. Would disease status modify the correlation between air pollution and FeNO? Would diagnosis of respiratory diseases indicate overall worse environmental quality (air, water, soil pollutants, less green space etc.) or worse healthcare in the neighborhood, which can confound the correlation between air pollution and FeNO?

Response 2: Thank you very much for your comment. We took into account the distance of schools from the measuring station in linear regression, and the results are presented in the table in supplementary materials (Table S1). After considering the additional feature, the direction, strength, and significance of the relationships studied did not change. All children included in the study lived within the city of Krakow. The inhabitants of Krakow have similar access to medical care (there are several pediatric wards located in several parts of the city in the city). Krakow's water supply and sewage system are of very good quality and are available to all residents.

Point 3: Discussion is needed regarding the impact of exposure misclassification on the observed correlations, e.g., exposures at home. Consider reporting the distances between the nearest monitors and the studied schools.

Response 3: In the limitations of the research, we have supplemented the information on the influence of passive smoking on the level of FeNO. As we mentioned in the answer above, we supplemented the regression analysis with the inclusion of the feature, which is the distance between the school and the measuring station.

Point 4: It is unclear what is the goal for the analysis on the correlation of FeNO between Stage I and Stage II (section 3.4). Can the authors further elaborate?

Response 4: Thank you very much for your comment. We removed this section from the results.

Point 5: Can the authors comment on the clinical significance of the results? Based on the linear regression results, it seems that the increase in FeNO was less than 1% for an SD increase in air pollutants. Given the range of FeNO in the population and in each clinical category, can the authors comment on the clinical significance of the findings?

Response 5: Thank you for your comment. The study we conducted is a population study, but we included a paragraph referring to the clinical importance of measuring FeNO in the pediatric population in the discussion.

Point 6: The authors listed the sample size of many previous studies on this topic in the introduction. If these studies showed consistent results, small sample size might not be a sufficient reason for an additional study with a larger sample size, especially if other studies with smaller sample size had more information on potentially important confounders. Can the authors elaborate the gaps in the literature and what is the contribution of this study?

Response 6: Thank you very much for your comment. We tried to justify the purpose and validity of the research undertaken in more detail.

All the authors are unanimous in term of accepting above-mentioned changes. We hope that the version submitted will meet the criteria to be issued in the International Journal of Environmental Research and Public Health.

Thank you

Reviewer 2 Report

This is a well-prepared study and well-written paper. There is a couple of issues of concern that should be addressed prior to publication. First, since summer measurements are included, authors should explain why ozone, the primary air pollutant in the summer that can cause irreversible damage and restructure of lungs in children. If data availability is an issue that should be reported, and the lack of ozone should be explained in the discussion. In Data Analysis, the authors wrote that regression analysis was done to delineate the effect of age and gender; however, no results are included. Please eliminate the statement from Data Analysis. And include it a discussion in the limitations of the study. If authors elect to include the new results, the manuscript may be reviewed again.

Round 2

Reviewer 1 Report

I was probably not clear on my last comments. The distance from the monitoring station to the schools is useful to evaluate potential misclassification of exposures. For example, if a school is 10 km away from the monitoring station and another school is 1 km away from the monitoring station, the chance of misclassification for the first school is much greater than the second school. Is there potential differential misclassification, which can bias the results? Or is there nondifferential misclassification? This is something that should be discussed.

I don't think the distance from station to the schools should be controlled in the models, because it is not a confounder.

Author Response

Respected Reviewer,

We wish to submit a revised manuscript “Relationship between air pollution and the concentration of nitric oxide in the exhaled air (FeNO) in 8-9-year-old school children in Krakow” (Manuscript ID: ijerph-1226873) for further consideration by the International Journal of Environmental Research and Public Health.

We do confirm that we have taken all the valuable reviewers comments under consideration and applied changes to the manuscript. The improvements are marked in green in the manuscript.

Point 1: I was probably not clear on my last comments. The distance from the monitoring station to the schools is useful to evaluate potential misclassification of exposures. For example, if a school is 10 km away from the monitoring station and another school is 1 km away from the monitoring station, the chance of misclassification for the first school is much greater than the second school. Is there potential differential misclassification, which can bias the results? Or is there nondifferential misclassification? This is something that should be discussed.I don't think the distance from station to the schools should be controlled in the models, because it is not a confounder

Response 1: Respected Reviewer, Thank you very much for your review. We would like to apologise for not understanding your previous suggestion in full. In the manuscript, we have removed the variable, that is the distance between the school and the measurement station from theregression models. In the „Discussion” part, in the limitations section, we have reworded the statements regarding determination of exposure levels to air pollution.

This manuscript is a resubmission of an earlier submission. The following is a list of the peer review reports and author responses from that submission.

Round 1

Reviewer 1 Report

In this manuscript, the authors report the results of a research project aimed to analyze the relationship between FeNO and the level of air pollution among primary school children in Krakow, to identify the relationship between air pollution and eosinophilic respiratory tract inflammation in children. An appropriate study design is not provided, since the study included both healthy children and those with respiratory diseases, including asthma, in which it is well known that NO production in the airway epithelium is increased. In addition, although FeNO may be an elegant tool for monitoring of environmental health effects of air pollution it is also affected by the prevalence of atopy in the studied population. In the absence of this important methodological information, the results of the study cannot be understood, given that it is not clear to which population group they refer.

Author Response

December 16, 2020

Respected Reviewer

We wish to submit a revised manuscript "Relationship between air pollution and the concentration of nitric oxide in the exhaled air (FeNO) in 8-9-year-old school children in Krakow" (Manuscript ID: jcm-1023215) for further consideration by the Journal of Clinical Medicine.

We do confirm that we have taken all the valuable reviewers’ comments under consideration and applied changes to the manuscript. The improvements are marked in the manuscript.

All the authors are unanimous in term of accepting above-mentioned changes. We hope that the version submitted will meet the criteria to be issued in the Journal of Clinical Medicine.

Yours faithfully,

Marta Czubaj - Kowal

Reviewer 2 Report

The current study aimed to assess the relationship between FeNO and air pollution in schoolchildren (8-9 years old) living in a highly polluted area of Poland. The manuscript is interesting, it was carried out on a quite large sample, and the authors correctly acknowledge the main study limitations (actually no information about potential confounder like asthma, allergy, and/or medication use). However, the manuscript would need a deep revision of organization and language style. Please find below some specific comments and suggestions for improving the manuscript.

Abstract

  1. Line 28: “2% of the result value”-> unclear.

Introduction

  1. Line 43: “most polluted in Europe” -> “most polluted ones in Europe”.
  2. Lines 44-45 and text remainder: please use subscripts for “10” and “2.5” (with a dot rather than a comma) in PM notation.
  3. Overall, the introduction would need to be reviewed to better highlight the study contribution: what knowledge does it add or improve? This aspect is indeed only briefly mentioned at the end of the section (Lines 76-78), while the study objectives were not even mentioned (at least not explicitly).
  4. Fifty quoted references may be too much for an introduction section. Despite well written, the long description of the possible underlying mechanisms (Lines 46 to 75) may be moved to the discussion. Conversely, the Authors may consider enlarging the description of previous reports on FeNO and air pollution, briefly highlighting the potential limitations that the current study is aimed to address.

Methods

  1. Line 100: “performed twice” -> do the Authors refers to the two seasonal measurements or to repeated consecutive measurement? In the latter case, which one was retained? The mean? Please clarify.
  2. Lines 106-108: the spatiotemporal resolution of the pollutants should be described more in detail; the reader will discover only later that FeNO measurements were correlated to air pollutants (from monitoring stations? at which spatial resolution?) on the day of measurement or on the week before (median).
  3. Line 106: “the concentration of air pollution in the particular matter...” -> please check this sentence.
  4. Please also explain that data analysis was carried out in two or three stages: autumn-winter, spring-summer, and the sub-analysis on the 4580 children performing two measurements.
  5. Line 110: how did the author choose to present the mean or the median for quantitative variables?
  6. Why did the Authors choose to use correlations rather than, for example, generalized linear models (for asymmetric outcomes)?
  7. Line 119: “accepted”-> “set”.

Results

  1. Line 125: “(50%)” -> please add the first decimal digits (to highlight the “slightly more”).
  2. Were other child characteristics recorded aside from age and gender, for example, height, weight, BMI? If so, a descriptive table should be added.
  3. Line 129: what is the p-value of 0.002 (please use the dot before decimal digits) for? Does it refer to FeNO comparison among October, November, December and January? If so, it should be clarified in the Table 1 legend.
  4. Line 130: “particular months” -> “specific months”.
  5. Line 130: “of the autumn-winter study” -> actually May and June are also shown in Table 1.
  6. Line 131: “shows Table 2”: please check the sentence.
  7. Line 133 and remainder of the text: is “R” for the Spearman’s rho? If so, please use rho.
  8. Lines 137-138 and remainder of the text: please uniform the format of the dates in the whole text and in Figures. Here they are in the dd-mm-yyyy format, while in figures they appear to be in the m-d-yy format.
  9. Line 143: please delete the dot at the end.
  10. Table 1: reporting a summary of the two whole seasons (as in Table 3) may help. Please clarify what is the p-value referred to.
  11. Table 2: why is spring-summer not there?
  12. Table 5: what is “**vs Table 1??** at the end of the legend?
  13. Table 7: adding the row-column totals would help a lot the reader. Please remind the reader what the p-value refers to in the legend or a footnote.
  14. Figures 1, 2 and 4: the Authors may consider removing these figures (especially Figure 2 that is just the combination of figures 3 and 5).

Discussion

  1. Line 198. “in large” -> “in a large”.
  2. Line 205. Why “interestingly”?
  3. Lines 198-216: this is just a repetition of the results, not a discussion. This part should rather emphasize the take-home message of the manuscript.
  4. Lines 217-218: unclear.
  5. Line 222: “assess” -> “have assessed”.
  6. Line 223: “what” -> “that”.
  7. Line 223: “most of this works was” -> “most of these works were”.
  8. Lines 223-224: this is an example of a sentence highlighting the study contribution (it may be moved or copied to the introduction).
  9. Lines 227-228: are these all the previous studies on the topic or only a few examples? Please consider focusing the discussion on the differences (study design, results) of the current study and the previous ones.
  10. Line 231: “were” -> “was”.
  11. Line 232: “were significant only” -> unclear sentence.
  12. Line 252: please check “1 211”.
  13. Line 260: “who in” -> please write the verb (“assessed...”) before saying where (“in the study...”); namely “who assessed....in a study....”.
  14. Line 267: “showed in” -> please say what was showed (“a relationship between...”) before saying where it was showed (“in a group...”); namely “showed a relationship...in a group...”.
  15. Please consider discussing more the differences (study design, results) of the current study and the previous ones. Although well written, the discussion is more similar to a literature review in the present form. Moreover, the Authors might consider removing the adult-elderly part.
  16. Line 297. Unclear, please check the sentence.

Conclusions

   43. Line 314. “2% of the result value”-> unclear.

Author Response

(The authors gave the same response as above.)

Reviewer 3 Report

General Comments:

In this manuscript, the authors investigate the association between air pollution and nitric oxide (FeNO) in exhaled air in school children in Krakow, Poland. The main finding of this study was that CO8h, C6H6, PM10, and NO were weakly but significantly correlated with FeNO in participants. The topic is interesting and important, but there are some severe flaws in the statistical analysis and interpretation of results. None of the associations were adjusted for confounding and causal language is used to describe study results. Additionally, the authors do not provide any demographic information on the study population beyond gender and age.

Major Comments:

  • I would highly recommend including a table of demographic information on the participants. Table would include number of participants in each phase, gender, ages, and any other demographic information.
  • Methods, line 86: Was any more information collected from the participants, E.g. household smoking, socioeconomic status, health behaviors (exercise), etc.? Without this information it is impossible to evaluate if the associations they found were due to differences in air pollution levels or rather due to differences in other covariates, e.g. smoking or socioeconomic status (see also my next comment).
  • Methods, line 115: Non-parametric t-test like the Mann-Whitney U test were done instead of a linear regression adjusting for confounders. Not adjusting for confounding (e.g. smoking or socioeconomic status) may create biased estimates. Therefore, their current results may potentially be explained by differences in other factors (e.g. smoking or socioeconomic status), for which they did not account in their analyses.
  • Methods, line 114: FeNO concentration was divided into groups, is there clinical significance for these groups?
  • Methods, line 106: It is unclear how the pollutants were assigned to the study participants. Based on their residential addresses? And how was the resolution? If it's data from monitoring stations, how many were there? Was there enough spatial variation in exposure levels?
  • Results, line 122: Were the Phase II measurements done on the same children as phase 1?
  • Results, line132 & line 153: Is the 1ppb median difference between boys and girls clinically significant?
  • Results, line 143 & line 159: Why did the authors use 7 day average air pollution estimates? This needs to be explained in the methods section.
  • Discussion, line 201: all correlation coefficients were very weak (R<0.15). Just because these correlations are statistically significant does not mean they are clinically significant or causal. Especially since all correlations are cross sectional and not adjusted for confounding.
  • Discussion line 207: Why was it expected to see weaker relationships between FeNO and air pollutants in the spring?
  • Discussion lines 217 - 294: did these other studies that saw a correlation adjust for confounding? What were their correlation coefficients and were they larger or similar to the ones found in this study? Are the methods used in these other studies similar to your study?
  • Discussion, line 295: How did your study ‘confirm’ that FeNO is a simple method for detecting airway inflammation?
  • Discussion, line 203: in this paragraph you state “The limitation of our work is that it was a one-center study of the population of 8-9 year old pupils. We had no information about respiratory diseases, atopy, inhalation allergy, asthma, medication use, exposure to tobacco smoke and other factors influencing FeNO measurement. These additional factors, as well as allergy to trees or grass pollens and possible asthma exacerbations probably had an impact on higher FeNO values during spring-summer in some of the examined children.” How do you think these systematic errors effected your results and do these limitations change the conclusions that you reach in your study?
  • Conclusions, line 313: “Although pollutants affect the measurement of FeNO, their degree of influence is relatively small and for individual pollutant agents, it reaches a maximum of 2% of the result value.” What does this mean and where did the 2% value come from?

Minor comments:

  • Page 2 line 50: the phrase “die of smog” is confusing.
  • Page3 line 86: What does “attending the third classes” mean?
  • Page 4 line163: highest decrease twice?
  • Page 8 line 201: R=01491 need to include decimal point
  • Page 10 line299: The phrase ‘homogenous in gender’ is in correct because both genders were included.
  • Include units in tables
  • Include what test was used to get the p-value in tables
  • Throughout: there is inconsistent use of ‘.’ and ‘, ‘for the decimal place marker

Author Response

(The authors gave the same response as above.)

Round 2

Reviewer 1 Report

I thank the authors for the effort they have made to improve the paper.
Despite this effort, the paper still presents serious problems in all sections: starting with the Introduction, which is very confusing and does not clearly define the reference context with the knowledge gaps that the paper intends to cover. Therefore, the research question is also not clearly stated. Subsequently, in the Material and Methods section, study design is missing entirely, and therefore it is unclear whether this study is a population-based study or there was a choice and how it was performed of the selected schools. Also in this section the methods are not detailed in subsections to describe the study site, environmental pollution, participants, procedures, etc. The results are confusing and as presented it is not possible to deal with the conclusions described by the authors.
The discussion is confused and too long; furthermore, the authors refer to some studies, which have a different study design, and therefore cannot be compared with the results of this study. It is inappropriate to include references to other age groups in the discussion, for example adults and the elderly.
Finally, even the conclusions do not respond to the aim of the study "The aim of the study was to demonstrate inflammation of the respiratory tract by determining the concentration of nitric oxide in the exhaled air (FeNO) in 8-9 year old children attending primary schools in KCrakcow and to determine the degree of correlation of these measurements with air pollution in the PM2.5, PM10, NO, C6H6 and CO in two time periods: spring-summer and autumn-winter ", since the authors conclude with this sentence "We suggest that the air pollution have weak impact on the clinical significance of the assessment of eosinophilic inflammation in the airways by FeNO.", but clinical aspects are not considered in this paper.

Author Response

Respected,

Reviewer

December 28, 2020

We wish to submit a revised manuscript "Relationship between air pollution and the concentration of nitric oxide in the exhaled air (FeNO) in 8-9-year-old school children in Krakow" (Manuscript ID: jcm-1023215) for further consideration by the Journal of Clinical Medicine.

Thank you very much for your review and suggestions for our manuscript, including them will certainly increase the level of the manuscript.

We do confirm that we have taken all the valuable reviewers comments under consideration and applied changes to the manuscript. The improvements are marked in blue in the manuscript and detailed in the table in the attached file.

Yours faithfully,

Marta Czubaj-Kowal

Reviewer 2 Report

Thank you for answering all of my comments.

Author Response

Respected,

Reviewer

December 28, 2020

Thank you very much for your opinion and all suggestions you gave us to make our manuscript better.

Yours faithfully,

Marta Czubaj-Kowal